# Optimal Central Frequency for Non-Contact Vital Sign Detection Using Monocycle UWB Radar

**DOI:** 10.3390/s20102916

**Published:** 2020-05-21

**Authors:** Artit Rittiplang, Pattarapong Phasukkit, Teerapong Orankitanun

**Affiliations:** 1Faculty of Engineering, King Mongkut’s Institute of Technology Ladkrabang, Bangkok 10520, Thailand; 59601306@kmitl.ac.th; 2Engineering Department, Royal Thai Naval Academy, Samutprakarn 10270, Thailand; teerapong.o@navy.mi.th

**Keywords:** UWB radar, monocycle pulse, non-contact vital sign detection, optimal frequency, Bessel-Gaussian integral, Hankel Transform

## Abstract

Ultra-wideband (UWB) radar has become a critical remote-sensing tool for non-contact vital sign detection such as emergency rescues, securities, and biomedicines. Theoretically, the magnitude of the received reflected signal is dependent on the central frequency of mono-pulse waveform used as the transmitted signal. The research is based on the hypothesis that the stronger the received reflected signals, the greater the detectability of life signals. In this paper, we derive a new formula to compute the optimal central frequency to obtain as maximum received reflect signal as possible over the frequency up to the lower range of Ka-band. The proposed formula can be applicable in the optimization of hardware for UWB life detection and non-contact monitoring of vital signs. Furthermore, the vital sign detection results obtained by the UWB radar over a range of central frequency have been compared to those of the former continuous (CW) radar to provide additional information regarding the advantages and disadvantages of each radar.

## 1. Introduction

Nowadays, radars not only are used to detect long-ranged targets such as military airplanes, but also they are used for short-range detections, especially in the monitoring of human activities, monitoring sleeping infants or adults, etc. In such biomedical applications, radars can be widely categorized into types, namely continuous wave (CW) radar and ultra-wideband (UWB) radar. Each of these radars has its own advantages and disadvantages as follows.

CW radars can provide accurate Doppler frequency measurements. They are of compact size, have low cost, longer detection range, and robust operation for practical portable and handheld applications. Therefore, CW radars are popular monitoring tools of human health such as sleep apnea syndrome, breathing disorder detection, and non-contact monitoring of the respiration of patients in hospitals [1,2,3,4,5,6,7,8,9,10,11].

UWB radars emit low electromagnetic radiation and, thus, consume relatively low levels of power. This is because UWB pulse duration is typically in the order of nanoseconds. UWB radars can provide high precision in target ranging up to centimeter scale, and have more penetrating capabilities through obstacles compared with CW radars. Nowadays, UWB radars are popularly used for detecting multiple targets through walls, finding individuals buried underneath earthquake rubble, breast cancer monitoring, medical microwave imaging, and non-contact vital sign detection [12,13,14,15,16,17,18,19,20,21,22,23,24,25,26,27,28,29,30,31,32,33,34,35,36,37,38,39].

Due to the distance between the target and the radar, a stronger received signal is strictly required to overcome noises and interferences, especially for tiny vital motions such as breathing and heartbeat motions. For UWB radars, the strengths of the heartbeat and breathing signals can be improved by adjusting the central frequency of the mono-pulse transmitted waveform. For CW radar, the carrier frequency of CW radars can be increased up to the lower region of the Ka-band to improve detection accuracy, because the short wavelength of the Ka-band sensor is capable of detecting small movements like chest-wall motion [7,8]. In practice, we do not have the luxury of unlimited bandwidth for which we can sweep across the entire range of frequencies. This is because the remote-sensing hardware (like antennas, PA, LNA), once set up, can operate at best on a few gigahertz of bandwidth. This practical problem has become our motivation to investigate the optimal central frequency for non-contact vital sign detection using a monocycle UWB radar. The organization of this paper is as follows.

The theoretical background is presented in Section 2. Then, in Section 3, a new model is derived from approximation techniques to theoretically investigate the optimal frequency, key parameters, behaviors and recovering vital sign spectra. The formula to obtain the optimal central frequency is derived from the new model into the simple formula. The simulations and experiments are presented in Section 4 and Section 5, respectively. As an extra study, in Section 4.4, the central frequency of the UWB radar has been compared to the carrier frequency of CW radar to provide additional information regarding the advantages and disadvantages of non-contact vital sign detection.

## 2. Background

### 2.1. Transmitted Signal Model

In impulse radio ultra-wideband (IR-UWB) radar applications, one of the most simple-to-generate transmitting pulses is monocycle waveform, as illustrated in Figure 1a. The waveform defined as the first derivative of the Gaussian pulse can be easily generated by microwave circuits or FPGA boards with low distortion [14,15,16,17,18,19,20,21,22,23,24,25]. In time domain, the amplitude, *s*(*t*), of the monocycle waveform is defined as [20,21,22]
(1)$$s\left(t\right)=-\frac{2\sqrt{e}}{t_{d}}A_{0}te^{-2{\left(\frac{t}{t_{d}}\right)}^{2}}$$
where *t_d_* = 1/(π*k*_c_) is the time duration between the maximum and minimum, *k*_c_ is the central frequency (Hz), and *A*_0_ is the transmitting amplitude. The maximum and minimum values of *s*(*t*) are, respectively, *A*_0_ and −*A*_0_ due to the multiplication factor −2√e/*t_d_*.

In the frequency domain, the monocycle wave form of (1) can be obtained by applying the Fourier transform (note that *tf*(*t*)
↔
(*j*/2π) (*dF*(*k*)/d*k*))
(2)$$S\left(k\right)=\overset{\infty }{\underset{-\infty }{\int }}s\left(t\right)e^{-j2\pi kt}dt=-\frac{2\sqrt{e}}{t_{d}}A_{0}\overset{\infty }{\underset{-\infty }{\int }}te^{-2{\left(\frac{t\;}{t_{d}}\right)}^{2}}e^{-j2\pi kt}dt=s_{0}ke^{-\alpha k^{2}}$$
where
(3)$$s_{0}=j\frac{A_{0}}{k_{c}^{2}}\sqrt{\frac{e}{2\pi }}\;\;\;\mathrm{and}\;\;\;\;\alpha =\frac{1}{2k_{c}^{2}}$$

The frequency domain of the transmitting signal is plotted in Figure 1b. From (2) and (3), the 3db-bandwidth of the transmitting signal *s*(*t*) depends on the central frequency *k_c_* as well as the order of the Gaussian pulse [20]. 

### 2.2. Received Signal Model

In general, the received signal can be modeled as the summation of multipath reflected signals from different objects with time delay, that is
(4)$$R\left(t,\tau \right)=\underset{p}{\sum }\sigma_{p}s\left(t-t_{p}\right)+\underset{o}{\sum }\sigma_{o}s\left(t-t_{o}\left(\tau \right)\right)+\underset{v}{\sum }\sigma_{v}s\left(t-t_{v}\left(\tau \right)\right)$$

The first term represents the multipath signals reflected from stationary objects with reflection coefficient (*σ_p_*) and time delay (*t_p_*). Also, the second term also represents the multipath signals reflected from non-stationary objects with reflection coefficient (*σ_o_*) and time delay (*t*_o_). Lastly, the third term is the multipath signals reflected from human vital signs with reflection coefficient (*σ_v_*) and time delay (*t_v_*). From (4), *R*(*t*,*τ*) is a function of “fast time” *t*, which is the wave-propagation time, and “slow time” *τ*, which is the time of signal detection.

For this research, the analysis focused on vital sign motion of one human. Therefore, the received signal in (4) can be reduced to
(5)$$R\left(t,\tau \right)=\sigma_{v}s\left(t-t_{v}\left(\tau \right)\right)$$
where the delay time *t_v_*(*τ*) is equal to *2d*(*τ*)/*c* (*c* is the speed of light in vacuum). The distance to the target is
(6)$$d\left(\tau \right)=d_{0}+m_{h}\sin\left(2\pi f_{h}\tau \right)+m_{r}\sin\left(2\pi f_{r}\tau \right)$$
where *d*_0_ is the nominal distance between the radar and the human chest. The *m_h_* and *m_r_* are, respectively, the movement amplitudes of the heart and respiration. The *f_h_* and *f_r_* are, respectively, the fundamental frequencies corresponding to heartbeat and respiration. From (6), the time delay in (5) can be written as
(7)$$t_{v}\left(\tau \right)=2d\left(\tau \right)/c=t_{0}+t_{h}\sin\left(2\pi f_{h}\tau \right)+t_{r}\sin\left(2\pi f_{r}\tau \right)$$
where *t*_0_, *t_r_*, and *t_h_* are the delays related to the human distance, respiratory and heart motions, respectively. 

### 2.3. Detection Block Diagram

A block diagram of the radar system and received signal in both fast and slow times are shown in Figure 2a,b, respectively. The principle of detecting physiological movements is based on the phase shift of the received reflected signals in the slow-time domain. The signals from stationary objects do not shift in the slow-time domain. The human chest exhibits periodic movement, as shown by (6) and (7), and will induce the shift of the signal along the slow-time domain. From [22,23,24,25,26,27,28,29,30,31,32,33,34,35,36,37,38,39], the vital sign spectrum *Y*(*t*_0_,*f*) is defined as the Fourier transform of the received signal in its slow-time domain that is
(8)$$Y\left(t_{0},f\right)=\sigma_{v}\overset{\infty }{\underset{u=-\infty }{\sum }}\overset{\infty }{\underset{l=-\infty }{\sum }}C_{lu}\left(t_{0}\right)\delta \left(f-lf_{h}-uf_{r}\right)$$
where
(9)$$C_{lu}\left(t_{0}\right)=\int_{-\infty }^{\infty }J_{l}\left(\frac{4\pi m_{h}}{\lambda }\right)J_{u}\left(\frac{4\pi m_{r}}{\lambda }\right)S\left(k\right)dk$$
defined as the spectral coefficient of the *l*-th harmonic of the heartbeat and the *u*-th harmonic of respiration. In (8), the *δ*(.) is a Dirac delta function. In (9), *J_l_* and *J_u_* are Bessel functions of the first kind for heartbeat and respiration, respectively. The *S*(*k*) is defined in (2) and λ is the wavelength associated with the central frequency of transmitted wave. From (9), it is seen that the spectral coefficient *C_lu_* is dependent on both the central frequency (through λ) of the signal and the movement amplitude, *m_r_* and *m_h_*. The coefficient *C_lu_* not only is related to the harmonics of respiration and heartbeat signals, but also related the intermodulation effects between them as well [8]. 

## 3. Spectral Coefficient Solution 

In this section, the spectral coefficients *C_lu_* in (9) will be simplified using an approximation technique. Then the formula of the optimal central frequency will be derived by taking the derivative of the magnitude of the received signal with respect to the central frequency of the transmitted signal. The optimization criteria aim to maximize the magnitude of the received signal. 

In general, the heart movement is usually less than 0.3 mm [40]. For frequencies up to 40 GHz, the Bessel functions *J_l_* in (9) can be approximated [7], that is
(10)$$J_{l}\left(\frac{4\pi m_{h}}{\lambda }\right)\approx \frac{1}{l!}{\left(\frac{2\pi m_{h}}{\lambda }\right)}^{l}$$

Figure 3 shows the comparison between the original Bessel function of the first kind and its linear approximation for (a) heartbeat, and (b) respiration. However, for respiration, the movement amplitude *m_r_* is large compared to the wavelength *λ*. The approximation does not provide good accuracy, as shown in Figure 3b. For the respiration case, we use
(11)$$J_{u}\left(\frac{4\pi m_{r}}{\lambda }\right)=\overset{\infty }{\underset{p=0}{\sum }}\frac{{\left(-1\right)}^{p}{\left(2\pi m_{r}/\lambda \right)}^{u+2p}}{p!\left(u+p\right)!}$$

Next, by substituting (2), (10) and (11) into (9) and letting λ = *c/k,* we have
(12)$$C_{lu}\left(t_{0}\right)=\frac{s_{0}}{l!}{\left(\frac{2\pi m_{h}}{c}\right)}^{l}\overset{\infty }{\underset{p=0}{\sum }}\frac{{\left(-1\right)}^{p}{\left(2\pi m_{r}/c\right)}^{u+2p}}{p!\left(u+p\right)!}\int_{-\infty }^{\infty }k^{u+l+2p+1}e^{-\alpha k^{2}}dk$$

The integral term of the above equation can be evaluated by applying the Gaussian integral of general order [41], that is
(13)$$\begin{array}{l} C_{lu}\left(t_{0}\right)\\ =\{\begin{array}{ccc} \frac{s_{0}}{l!}{\left(\frac{2\pi m_{h}}{c}\right)}^{l}\sum_{p=0}^{\infty }\frac{{\left(-1\right)}^{p}{\left(\frac{2\pi m_{r}}{c}\right)}^{u+2p}}{p!\left(u+p\right)!}\sqrt{\frac{\pi }{\alpha }}\frac{\left(n-1\right)!!}{{\left(2\alpha \right)}^{\frac{n}{2}}} & ; & \mathrm{for}\;n=u+l+2p+1=\mathrm{even}\\ 0 & ; & \mathrm{for}\;n=u+l+2p+1=\mathrm{odd} \end{array} \end{array}$$

By letting α and *s*_0_ as in (3) and *k_c_* = c/*λ_c_*, we have
(14)$$C_{lu}\left(t_{0}\right)=\frac{jA_{0}\sqrt{e}}{l!}{\left(\frac{2\pi m_{h}}{\lambda_{c}}\right)}^{l}\overset{\infty }{\underset{p=0}{\sum }}\frac{{\left(-1\right)}^{p}{\left(2\pi m_{r}/\lambda_{c}\right)}^{u+2p}}{p!\left(u+p\right)!}(u+l+2p)!!$$

The power series term above can be viewed as the Bessel function of (11) multiplied by the double factorial term (*u + l + 2p*)!! Thus, we define
(15)$$G_{lu}\left(\frac{4\pi m_{r}}{\lambda_{c}}\right)=\overset{\infty }{\underset{p=0}{\sum }}\frac{{\left(-1\right)}^{p}{\left(2\pi m_{r}/\lambda_{c}\right)}^{u+2p}}{p!\left(u+p\right)!}(u+l+2p)!!$$
where (*u + l +* 2*p*)!! = (*u + l +* 2*p*)!/[2^(*u+l*+2*p*−1)/2^.((*u + l +* 2*p* − 1)/2)!], the above function depends on 4π*m_r_*/λ_c_ with *l* and *u* orders, and it is convergent by the well-known ratio test
(16)$$\underset{p\to \infty }{\lim}\left|\frac{a_{p+1}}{a_{p}}\right|=\underset{p\to \infty }{\lim}\frac{\left(u+l+2p\right){\left(2\pi m_{r}/\lambda_{c}\right)}^{2}}{\left(p+1\right)\left(u+p+1\right)}=0$$

Consequently, (14) can be rewritten as
(17)$$C_{lu}\left(\lambda_{c},m_{r},m_{h}\right)=\{\begin{array}{ccc} \frac{jA_{0}\sqrt{e}}{l!}{\left(\frac{2\pi m_{h}}{\lambda_{c}}\right)}^{l}G_{lu}\left(\frac{4\pi m_{r}}{\lambda_{c}}\right) & , & u+l=\mathrm{odd}\\ 0 & , & u+l=\mathrm{even} \end{array}$$

From the above equation, we can initially observe that for given values of *m_r_* and *m_h_*, the coefficient *C_lu_* (the received signal strength) can be improved by increasing the central frequency (*k*_c_ = c/*λ*_c_). Note also that the −3 db bandwidth is dependent on the central frequency as well as the type of Gaussian pulse defining the transmitting signal [20]. In the next section, the optimal central frequencies for both respiratory and heartbeat models will be derived.

### 3.1. Respiratory Model

The harmonic components of the respiratory can be obtained by letting *l* = 0 [25], in (17) that is
(18)$$C_{0u}\left(\lambda_{c},m_{r}\right)=\{\begin{array}{ccc} jA_{0}\sqrt{e}G_{0u}(\frac{4\pi m_{r}}{\lambda_{c}}) & , & u=\mathrm{odd}\\ 0 & , & u=\mathrm{even} \end{array}$$

The spectral coefficient decays as the order number of *u* increases. From [25,26,27], the harmonic orders of *u* = 0, 1, 2 and 3 are significant for respiration spectrum, and we can evaluate them as
(19)$$\begin{array}{c} C_{00}\left(\lambda_{c},m_{r}\right)=C_{02}\left(\lambda_{c},m_{r}\right)=0\\ C_{01}\left(\lambda_{c},m_{r}\right)=jA_{0}\sqrt{e}G_{01}\left(\frac{4\pi m_{r}}{\lambda_{c}}\right)\\ C_{03}\left(\lambda_{c},m_{r}\right)=jA_{0}\sqrt{e}G_{03}\left(\frac{4\pi m_{r}}{\lambda_{c}}\right) \end{array}$$

By substituting (19) into (8), we have
(20)$$Y_{resp}\left(t_{0},f\right)=\sigma_{v}\overset{3}{\underset{u=0}{\sum }}C_{0u}\left(\lambda_{c},m_{r}\right)\delta \left(f-uf_{r}\right)=R_{1}\delta \left(f-f_{r}\right)+R_{3}\delta \left(f-3f_{r}\right)$$
where *R*_1_ is the coefficient of the fundamental frequency *f_r_*. *R*_3_ is the coefficient of the third harmonic 3*f_r_*. These quantities *R*_1_ and *R*_3_ can be evaluated as
(21)$$R_{1}=j\sigma_{v}A_{0}\sqrt{e}G_{01}\left(\frac{4\pi m_{r}}{\lambda_{c}}\right)\;\;\;\;\mathrm{and}\;\;\;R_{3}=j\sigma_{v}A_{0}\sqrt{e}G_{03}\left(\frac{4\pi m_{r}}{\lambda_{c}}\right)$$

Note that the *R*_3_ should be minimized as possible, since it could interfere with the detectability of the fundamental frequency of the heartbeat signal. This is because, while their spectra are very close, the amplitude of R_3_, which is relatively large, could superimpose on the heartbeat signal [25]. To improve the detectability of the fundamental respiration frequency, the optimal frequency can be obtained by taking the derivative of the above equation with respect to central frequency, that is
(22)$$\frac{\partial R_{1}}{\partial k_{c}}=\frac{\partial G_{01}\left(4\pi m_{r}/\lambda_{c}\right)}{\partial k_{c}}=0$$

*G*_01_ was computed from (15) power-series expansion, thus we have
(23)$$\overset{\infty }{\underset{p=0}{\sum }}\frac{{\left(-1\right)}^{p}\left(2p+1\right)\left(2p+1\right)!!}{p!\left(p+1\right)!}{\left(\frac{2\pi m_{r}}{\lambda_{c}}\right)}^{2p}=0$$
(24)$$\begin{array}{c} 1-\frac{3\left(3\right)!!}{1!2!}{\left(\frac{2\pi m_{r}}{\lambda_{c}}\right)}^{2}+\frac{5\left(5\right)!!}{2!3!}{\left(\frac{2\pi m_{r}}{\lambda_{c}}\right)}^{4}-\frac{7\left(7\right)!!}{3!4!}{\left(\frac{2\pi m_{r}}{\lambda_{c}}\right)}^{6}+\frac{9\left(9\right)!!}{4!5!}{\left(\frac{2\pi m_{r}}{\lambda_{c}}\right)}^{8}\\ \;\;\;\;\;\;\;\;-\frac{11\left(11\right)!!}{5!6!}{\left(\frac{2\pi m_{r}}{\lambda_{c}}\right)}^{10}+\frac{13\left(13\right)!!}{6!7!}{\left(\frac{2\pi m_{r}}{\lambda_{c}}\right)}^{12}-\frac{15\left(15\right)!!}{7!8!}{\left(\frac{2\pi m_{r}}{\lambda_{c}}\right)}^{14}+\ldots =0 \end{array}$$

The high-order polynomial cannot easily be factored, we need to use numerical techniques to find a polynomial’s roots. With the polynomial roots command in MATLAB program and for-loop coding, we discovered that 2π*m_r_*/*λ_c_* = 0.597 is the valid root corresponding to the global maximum of the function. Finally, the optimal central frequency (*k_OR_*_1_) to maximize the respiration strength can be derived as
(25)$$k_{OR1}=0.597\frac{c}{2\pi m_{r}}$$
which is inversely proportional to the respiratory movement amplitude *m_r_*. In practice, for a fixed or approximated value of *m_r_*, the corresponding optimal central frequency of the transmitted signal can be easily calculated by using (25).

### 3.2. Heartbeat Model

Similar to the respiration case, the harmonics of the heartbeat can be obtained by substituting index *u* = 0 into (17). We have
(26)$$C_{l0}\left(\lambda_{c},m_{r},m_{h}\right)=\{\begin{array}{ccc} \frac{jA_{0}\sqrt{e}}{l!}{\left(\frac{2\pi m_{h}}{\lambda_{c}}\right)}^{l}G_{l0}\left(\frac{4\pi m_{r}}{\lambda_{c}}\right) & , & l=\mathrm{odd}\\ 0 & , & l=\mathrm{even} \end{array}$$

It is seen from the above equation that *C_l_*_0_ is dependent on *k*_c_, *m_h_* and especially *m_r_*. This is because the chest motion *m_r_* also affects the heartbeat movement [8,24,25], The dominant spectra of the heartbeat signal are
(27)$$Y_{heart}\left(t_{0},f\right)=\sigma_{v}\overset{3}{\underset{l=0}{\sum }}C_{l0}\left(\lambda_{c},m_{r},m_{h}\right)\delta \left(f-lf_{h}\right)=H_{1}\delta \left(f-f_{h}\right)+H_{3}\delta \left(f-3f_{h}\right)$$
where *H*_1_ is the coefficient of the fundamental frequency and *H*_3_ is the coefficient of the third harmonic of heartbeat, which can be written as
(28)$$H_{1}=j\sigma_{v}A_{0}\sqrt{e}\left(\frac{2\pi m_{h}}{\lambda_{c}}\right)G_{10}\left(\frac{4\pi m_{r}}{\lambda_{c}}\right)\;\;\;\;\;\;\mathrm{and}\;\;\;\;\;H_{3}=j\sigma_{v}A_{0}\sqrt{e}\frac{1}{6}{\left(\frac{2\pi m_{h}}{\lambda_{c}}\right)}^{3}G_{30}\left(\frac{4\pi m_{r}}{\lambda_{c}}\right)$$

The heartbeat estimation observed from the fundamental frequency can be erroneous; consequently, we can better detect that by maximizing its strength with the optimal frequency. We take the maximum derivative with respect to *k_c_*, that is
(29)$$\frac{\partial H_{1}}{\partial k_{c}}=\frac{\partial \left\{\left(\frac{2\pi m_{h}}{\lambda_{c}}\right)G_{10}\left(\frac{4\pi m_{r}}{\lambda_{c}}\right)\right\}}{\partial k_{c}}=0$$
and apply the product rule
(30)$$\left(\frac{2\pi m_{h}}{\lambda_{c}}\right)\frac{\partial G_{10}\left(\frac{4\pi m_{r}}{\lambda_{c}}\right)}{\partial k_{c}}+G_{10}\left(\frac{4\pi m_{r}}{\lambda_{c}}\right)\frac{\partial \left(\frac{2\pi m_{h}}{\lambda_{c}}\right)}{\partial k_{c}}=0$$

Then, substituting (15) for *G*_10_(4πm_r_/λ_c_) to the above equation yields
(31)$$\overset{\infty }{\underset{p=0}{\sum }}\frac{{\left(-1\right)}^{p}\left(2p+1\right)\left(2p+1\right)!!}{{\left(p!\right)}^{2}}{\left(\frac{2\pi m_{r}}{\lambda_{c}}\right)}^{2p}=0$$
(32)$$\begin{array}{c} 1-\frac{3\left(3\right)!!}{{\left(1!\right)}^{2}}{\left(\frac{2\pi m_{r}}{\lambda_{c}}\right)}^{2}+\frac{5\left(5\right)!!}{{\left(2!\right)}^{2}}{\left(\frac{2\pi m_{r}}{\lambda_{c}}\right)}^{4}-\frac{7\left(7\right)!!}{{\left(3!\right)}^{2}}{\left(\frac{2\pi m_{r}}{\lambda_{c}}\right)}^{6}+\frac{9\left(9\right)!!}{{\left(4!\right)}^{2}}{\left(\frac{2\pi m_{r}}{\lambda_{c}}\right)}^{8}\\ \;\;\;\;\;\;\;\;\;-\frac{11\left(11\right)!!}{{\left(5!\right)}^{2}}{\left(\frac{2\pi m_{r}}{\lambda_{c}}\right)}^{10}+\frac{13\left(13\right)!!}{{\left(6!\right)}^{2}}{\left(\frac{2\pi m_{r}}{\lambda_{c}}\right)}^{12}-\frac{15\left(15\right)!!}{{\left(7!\right)}^{2}}{\left(\frac{2\pi m_{r}}{\lambda_{c}}\right)}^{14}+\ldots =0 \end{array}$$

The valid root for maximization is 2π*m_r_/λ_c_* = 0.3901 with the same method as (24), then the optimal central frequency (*k_OH_*_1_) to maximize the heartbeat strength *H*_1_ is
(33)$$k_{OH1}=0.3901\frac{c}{2\pi m_{r}}$$

From the above equation, *k_OH_*_1_ is inversely proportional to *m_r_*. It is very important to note that although the *H*_1_ value can be maximized via the optimal central frequency as in (33), the undesired third respiratory harmonic (*R*_3_) begins to increase as the frequency increases, and *R*_3_ degrades the detection accuracy of the heartbeat signal. Hence, the optimal central frequency for increasing *H*_1_ strength should not give rise to *R*_3_. The relevant discussion is provided in Section 4.2.

### 3.3. Avoiding the Central Frequency for the Peak R_3_ harmonic and the Null Point H_1_

As mentioned before, the third respiratory harmonic, *R*_3_ will degrade the accuracy of detection and its value increases as the frequency increases. The constraint should put the range of frequency at which the radar operates to avoid the occurrence of the third harmonic, especially its peak value. Similar to the *R*_1_ case, the frequency at the maximum *R*_3_ can be obtained via derivatives,
(34)$$k_{avoid,R3}=1.1405\frac{c}{2\pi m_{r}}$$

Also, it is worth pointing out that there is one frequency at which *H*_1_ = 0. This is defined as the null point *k_null, H_*_1_, which can be obtained by solving
(35)$$H_{1}=\left(\frac{2\pi m_{h}}{\lambda_{c}}\right)G_{10}\left(\frac{4\pi m_{r}}{\lambda_{c}}\right)=0$$
the null frequency can be obtained as
(36)$$k_{null,H1}=0.9533\frac{c}{2\pi m_{r}}$$

On the contrary, the *R*_1_ strength has no null-point, which will be discussed in Section 4.1.

### 3.4. Vital Sign Spectrum Model of Monocycle UWB

According to (20) and (27), the vital sign signal model in (8) can be rewritten as
(37)$$\begin{array}{c} Y\left(t_{0},f\right)=j\sigma_{v}A_{0}\sqrt{e}\overset{Y_{resp}\left(t_{0},f\right)}{\overbrace{[G_{01}\left(\frac{4\pi m_{r}}{\lambda_{c}}\right)\delta \left(f-f_{r}\right)+G_{03}\left(\frac{4\pi m_{r}}{\lambda_{c}}\right)\delta \left(f-3f_{r}\right)}}\\ \underset{Y_{heart}\left(t_{0},f\right)}{\underbrace{+\left(\frac{2\pi m_{h}}{\lambda_{c}}\right)G_{10}\left(\frac{4\pi m_{r}}{\lambda_{c}}\right)\delta \left(f-f_{h}\right)+\frac{1}{6}{\left(\frac{2\pi m_{h}}{\lambda_{c}}\right)}^{3}G_{30}\left(\frac{4\pi m_{r}}{\lambda_{c}}\right)\delta \left(f-3f_{h}\right)]}}+Y_{inter}\left(t_{0},f\right) \end{array}$$
where
(38)$$Y_{inter}\left(t_{0},f\right)=\sigma_{v}\overset{\infty }{\underset{\begin{array}{c} u=-\infty \\ u\neq 0 \end{array}}{\sum }}\overset{\infty }{\underset{\begin{array}{c} l=-\infty \\ l\neq 0 \end{array}}{\sum }}C_{lu}\left(\lambda_{c},m_{r},m_{h}\right)\delta \left(f-lf_{h}-uf_{r}\right)$$

The *Y_inter_* is an intermodulation model caused by heartbeat and respiratory signals, where *C_lu_* is denoted in (17). The null-point value of *Y_inter_* occurs every quarter of the wavelength that is a point in which the quantity is zero; to avoid this, methods based on IQ detection, demodulation and antenna diversity have been proposed. The methods based on IQ demodulation were applied for accurately detecting low signal-to-noise ratio [30] and have the problem of unbalance between IQ channels, but theoretically, there is not intermodulation because it detects the phase. 

The proposed model (37) in the neglected intermodulation case is compared with the MATLAB simulation as shown in Figure 4, where the related parameters are referred from Table 1. Respiration and heartbeat movement amplitudes are given as 1.8 and 0.08 mm, respectively, for a relaxed human [8], and the central frequency is done with 6 GHz following the FCC mask.

As illustrated in Figure 4, the result from the proposed model is in good agreement with that of the simulation, while the proposed model is faster than computing the simulation. The intermodulation effect (Int08) and third respiratory harmonic (*R*_3_) occur at 0.8 and 1.2 Hz, respectively, and they are strongly dependent on the respiratory movements (chest motions). The third respiratory harmonic is a serious problem that is often close to the heartbeat frequency H_1_ [25], in this case, *H*_1_ is at 1.1 Hz. However, these interferences could be filtered out, as suggested in [25], or trying to find a best orientation, as proposed in [42]. 

## 4. Analysis of Optimal Central Frequency

In the following section, numerical simulations are performed to verify the performance of the new formula and optimal frequency.

### 4.1. Detected Respiratory Strength

The study of frequency response of the respiratory model (21) is shown in Figure 5. Figure 5a illustrates the value of *R*_1_ (dB) versus the central frequency with *m_r_* varying from 0.8 to 12 mm while *m_h_* is assumed to be fixed at 0.08 mm, which is typical in ordinary people [8]. The other parameters are fixed, as shown in Table 1. Figure 5b shows the plot of optimal central frequency versus respiratory amplitude *m_r_* with *m_h_* varying from 0.02 to 0.2 mm. 

From Figure 5a, the curve graph described the tendency to respond at greater respiratory amplitude *m_r_* when the central frequency starts to match the system’s natural frequency (its resonant frequency) of the periodically chest vibration. It is seen that, for each *m_r_*, as the central frequency increases, the value of *R*_1_ increases and reaches the highest value (optimal), and then starts to decline. For large values of *m_r_*, the value of *R*_1_ increases to the highest point and declines more quickly than those with smaller *m_r_*. This can be explained as follows: the larger value of *m_r_* increases the speed of *J_u_*(4*π**m_r_*/*λ*). The plot of the simulated optimal frequency versus the respiratory amplitude *m_r_* is shown in Figure 5b compared to the simple formula *k_OR_**_1_* in (25). The results show that the *k_OR_**_1_* formula (solid line) is almost identical to the simulation results (marker), which remains almost unchanged over the varying the heartbeat amplitude *m_h_.* This means that the optimal frequency is strongly dependent on the respiratory amplitude *m_r_.* Also, when the respiratory amplitude becomes large, the optimal frequency for the maximization decreases.

### 4.2. Detected Heartbeat Strength

Following the same suggestions in Figure 5, the accuracy of the heartbeat model (28) is also compared with the simulation versus the central frequency as shown in Figure 6a. 

As shown in Figure 6a, as the central frequency increases, the heartbeat value *H*_1_ increases until it reaches the maximum value at the optimal central frequency. When the central frequency becomes too high, the strength *H*_1_ starts decreasing, especially for those with large values of *m_r_*. Intuitively, if the respiratory amplitude *m_r_* becomes large, it causes the *H**_1_* peak to drop quickly. This is because the chest surface vibration interferes directly with the heartbeat motion corresponding to the *H*_1_ function in (28). This means that the strength *H*_1_ depends on *m_h_* and especially *m_r_*. 

The optimal frequency simulations in Figure 6a are compared with the simple Equation (33) as shown in Figure 6b with the same conditions as Figure 5b. The optimal central frequency begins to reduce when the respiratory amplitude becomes large. The simulation results are almost unchanged with the heartbeat amplitude *m_h_*, this means that the optimal frequency value is strongly dependent on the respiratory amplitude *m_r_*.

#### 4.2.1. Comparison of Heartbeat Strength with R_3_ Harmonic

According to Figure 4, the *H*_1_ spectrum tends to be associated directly to the third respiration harmonic (*R*_3_). In order to improve the *H*_1_ strength, the relative strength *H*_1_**/***R*_3_ must be considered as in Figure 7a. It is shown that as the central frequency increases, the relative strength decreases, but not cause *H*_1_/*R*_3_ ≤ 1 for the detectable *H*_1_ spectrum [8].

#### 4.2.2. Comparison of Heartbeat Strength with Intermodulation

The *H*_1_ strength is compared with the intermodulation effect (Int08) at 0.8 Hz according to Figure 4, resulting in Figure 7b with the same suggestions in Figure 5. The relative strength *H*_1_/Int08 decreases when the central frequency becomes high, but also not causing *H**_1_*/Int08 ≤ 1 [8].

According to Figure 7a,b, the *R*_3_ effect is more serious than the intermodulation, because *R*_3_ is very close to *H*_1_ and the highest peak of the harmonics. Therefore, the *H*_1_ signal can be maximized in the premise that the *R*_3_ harmonic is not so large as to affect the detection accuracy [8]. In addition, a best strategy could be to filter these harmonics and interferences, as suggested in [25], or trying to find a best orientation, as proposed in [42].

### 4.3. Discussion on the Optimal Central Frequency

We consider the behaviors *R*_1_*, H*_1_*, H*_1_/Int08, and *R*_1_*/R*_3_ versus the same frequency as shown in Figure 8 with the respiration and heartbeat amplitudes of 1.8 and 0.08 mm, respectively, which are the typical values of a relaxed human [8]. Figure 8a shows a graph of *R*_1_ and *H*_1_ magnitudes and the relative strength of *H*_1_*/R*_3_, *H*_1_/Int08, *R*_1_*/R*_3_ are analyzed in Figure 8b. 

For improving the *H*_1_ strength with the condition of *H*_1_*/R*_3_ > 1, its optimal frequency has the lowest value shown in Figure 8b; in this case, it should not exceed 7.4 GHz. The *R*_3_ harmonic can be removed, and the frequency can be increased up to the maximum *H*_1_ at 10 GHz computed from (33), corresponding to Figure 8a. In addition, we can avoid the null-point *H*_1_ from evaluating (36). In this case, calculating the null-point frequency is approximately 24 GHz, corresponding to Figure 8b. Using through-the-wall radar, the respiratory signal could be enough to detect life, and also maximized using the optimal central frequency computed from the simple Equation (25). In this case, calculating the optimal frequency is about 16 GHz, corresponding to Figure 8a. 

Finally, Figure 8 shows that the central frequency should be limited to the lower region of the *Ka*-band for typical values of human chest wall movement [8].

### 4.4. Comparison between UWB and CW Radars

In [8], the received baseband signal of CW radars for the vital sign signal can be approximated as
(39)$$B\left(\tau \right)=B_{0}\cos\left[\frac{4\pi d\left(\tau \right)}{\lambda }+\theta \right]$$

Determining the amplitude, *B*_0_ is assumed as 1, the distance *d*(*τ*) is denoted in (6), the total phase shift *θ* is assumed as 90°, the wavelength *λ* is demonstrated from *f* = 500 MHz to 40 GHz, and the related parameters are referred from Table 1. The CW carrier frequency is compared with the UWB center frequency for the analysis of the behaviors *R*_1_, *H*_1_, *H*_1_*/R*_3_, and *H*_1_*/I08,* as shown in Figure 9, with the relaxed respiration *m_r_* = 0.8–3 mm and heartbeat *m_h_* = 0.08 mm [8,23,24,25,26,28]. Note that the CW simulation results using the MATLAB program are equal to the ADS results in [8].

As illustrated in Figure 9a, the UWB radar can detect the respiratory strength better than CW at the same frequency; this does not apply at the higher frequency. Unfortunately, from Figure 9d, the relative strength *H*_1_*/R*_3_ of UWB is less than CW at the same frequency, which means the CW radar is more efficient than UWB for detecting the heartbeat signal at the same frequency. Further, from Figure 9b, both UWB and CW radars find it difficult to detect the heartbeat signal using a lower frequency < 1 GHz.

## 5. Experimentation for Optimal Central Frequency of Respiration

The UWB radar system used for this experiment is shown in Figure 10. The experimental components are listed in Table 2.

Before the actual experiment, the preliminary step was done by measuring the respiratory signal of a person sitting 1 m from the UWB radar, as shown in Figure 10. A belt sensor was used to measure the person’s respiratory movement amplitude and the result is shown in Figure 11a,b. As shown in Figure 11a, the peak value *m_r_* of the sinusoidal waveform is not constant over the time of measurement. In practice, this alteration is always the case and should be expected to be seen, although the person is sitting still. From the belt measurement, the average value of *m_r_* is approximately 9.5 mm. As shown in Figure 11b, the frequency domain shows two dominant spectra at 0.38 and 0.42 Hz, which correspond to respiratory rates of 22.8–25.2 breaths per minute. To measure the respiration signal, a monocycle waveform was generated by the UWB source with the 3-GHz central frequency and time duration *t_d_* = 0.1 ns computed by *t_d_* = 1/(π*k*_c_). The measurements of the transmitted waveform by the oscilloscope and corresponding frequency domain are shown in Figure 12a.

In this system, the peak transmitted power is about −5 dBm and the operable frequency is from 2 to 5 GHz for all components. The system bandwidth does not fully follow the FCC regulations of 3.1–10.6 GHz, which is set to limit interference to existing communication systems only, but in this experiment the power on the chest surface does not exceed 10 W/m^2^ (permissible exposure limit), thus the electromagnetic radiation poses no safety threat [43]. The received reflected signal from the human subject was amplified by the LNA and sent to the oscilloscope. The signal on the oscilloscope was captured and transferred to a PC via the GPIB port interface and then the data were discretized by MATLAB program for further signal processing. The slow-time signals were captured approximately at the rate of 512 times in 80 s. For each slow-time measurement, the received signals were measured over the duration of 25 ns (fast time), which is then discretized to 7985 points. Figure 12b shows the received signals measured in fast-time domain with multiple overlapping slow-time measurements. Figure 12c shows the corresponding 2D-matrix (7985 × 512) of fast-time versus slow-time plots. The antenna coupling as well as the signal-to-noise ratio (SNR) can be improved by using basic filters to remove static signals, like linear least-squares, smooth filter, and bandpass filter for slow and fast-time domains [22,31,44]. The fast Fourier transform (FFT) technique was used to compute the Doppler shift of the vital sign signals [22,23,24,25,26,27,28,29,30,31,32,33,34,35,36,37,38,39]. The plots of Doppler shift are shown in Figure 13a where the theory plot has been compared with experimental UWB measurement. In this case, the plots show no intermodulation between heartbeat and respiration, due to the use of 3-GHz central frequency which is too low.

Next, the main experiments were carried out to investigate the frequency characteristic of the received signal magnitude. The received signal measurements were taken with three trials, with three different persons, and the central frequency of the transmitted signal was varied from 2 to 4 GHz. As shown in Figure 13b–d, the measurement results of the three cases are compared with the theory plot where *m_r_* is assumed to be 9.5 mm and the optimal central frequency is calculated by the proposed Equation (25) to be 3 GHz.

From Figure 13b,c, it is probably difficult to see that the maximum magnitudes do occur at 3.0 and 3.1 GHz, respectively, because the experimental curves are almost flat. However, as shown in Figure 13d, it is obvious to see the optimal frequency characteristic where the normalized magnitude of the received signal is increased from 0.9 (at 2.0 GHz) to maximum 1.0 (at 2.6 GHz) and then starts to decline at the higher central frequencies. The mismatches between the theory plots and experimental plots are mostly due to the fact that the amplitude of chest movement in practice is not periodic over the time. As opposed to (6), which assumes *m_r_* to be constant, the amplitude of human chest movement does not follow perfect sinusoidal form, as measured and shown in Figure 11a. Another reason could be the loss of symmetry of the monocycle pulse that normally comes from the limited bandwidth of the antennas and other RF components, which is very typical in the real-world situations [44]. 

The experiment was not carried out to measure the heartbeat signals due to testing the large respiratory amplitude movement (*m_r_* = 9.5 mm) and the lack of higher frequency capabilities of our hardware, these results correspond to the heartbeat strength analysis in Figure 6a. To see the better optimal frequency characteristic of the vital-sign signal strength, a wider range of frequencies for the experimental part may be suitable for future study. 

## 6. Conclusions

The detection of vital signs using radar system is essentially based on interpreting the echo signals scattered from human micromovements. These vital movements induce changes in frequency, amplitude and time-of-arrival of the received signals. Monocycle UWB radar is popularly used as a non-contact monitor since it could free the person from wearable sensors and imposes no infringement on personal privacy. However, due to the distance between the target and the radar, the stronger received signal is strictly required to overcome noises and interferences, especially for tiny vital motions such as breathing and heartbeat motions. In this paper, it was shown that, while other parameters were kept as constants, the magnitudes of the heartbeat and breathing signals could be improved by adjusting the central frequency of the transmitted monocycle waveform.

In this study, the formula for optimal central frequency had been derived and presented, based on the calculus derivative-optimization method. Note that, due to the greater intensity of breathing motions, it is easier to detect respiratory movement than heartbeat. For the respiratory detection, the frequency obtained from the proposed formula could be directly applied. However, the heartbeat signal is so weak and is always covered by the larger respiratory signal. Therefore, in order to obtain the optimal central frequency for the heartbeat signal, the proposed formula must be used in conjunction with the minimization of the third harmonic of the respiration signal.

The experimental results and formula are quite different because of the loss of symmetry of the monocycle pulse that normally comes from bandwidth antennas, hardware, environment, etc., in the real-world applications. However, the proposed formula to compute optimal central frequency can be useful in estimating the range of frequencies at which the remote-sensing hardware operates. According to Figure 5a, Figure 6a and Figure 7a, the optimal frequency is higher for smaller respiratory movement *m_r_*. To avoid significant performance deterioration as *m_r_* increases, we could estimate for optimizing the central frequency that should be limited to the lower region of the Ka-band.

Furthermore, as an extra study, we compared the performances of UWB and CW radars. According to simulation results, for given frequencies, the UWB radar is better off at detecting the respiratory signal, while the CW radar performs very well in detecting small signals from heartbeat motion. For frequencies lower than 1 GHz, both uses of UWB and CW radars find it difficult to detect a small heartbeat signal.

Lastly, it is worth pointing out that this study was done without considering the problems of sampling rates of both fast-time and slow-time domains. Also, the proposed analytical framework was based on the approximation of the Bessel function which was accurate for the heartbeat no greater than 0.3 mm. The aforementioned issues will be revisited and improved in future studies.

## Figures and Tables

**Figure 1 sensors-20-02916-f001:**
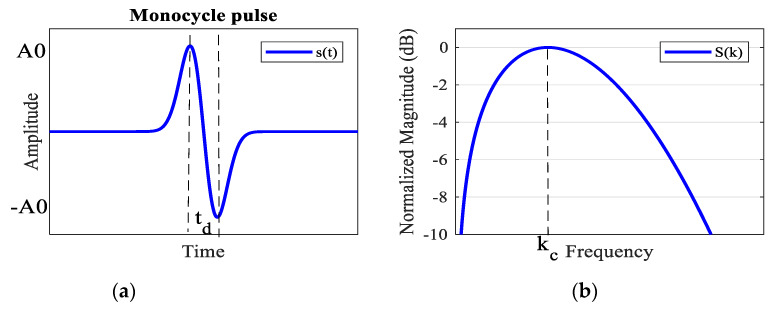
The 1st order Gaussian function: (**a**) time domain (**b**) frequency domain.

**Figure 2 sensors-20-02916-f002:**
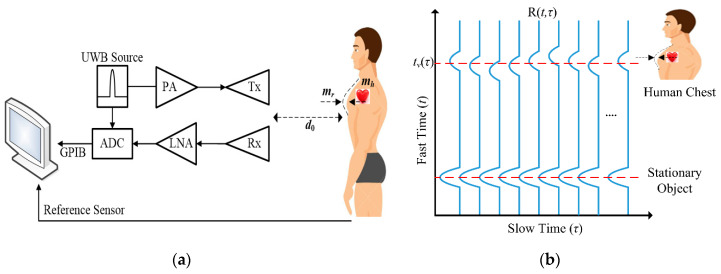
(**a**) A simplified block diagram of the radar system; (**b**) Rx signal depends on the chest movement.

**Figure 3 sensors-20-02916-f003:**
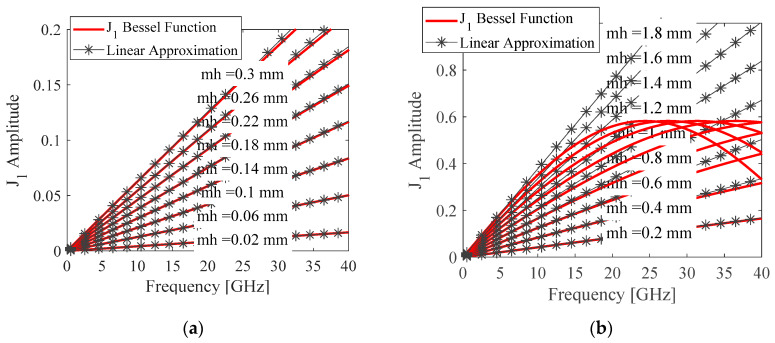
Comparison between the Bessel function of the first kind of order 1 and the linear approximation for the movement amplitude cases of (**a**) heartbeat; (**b**) respiration.

**Figure 4 sensors-20-02916-f004:**
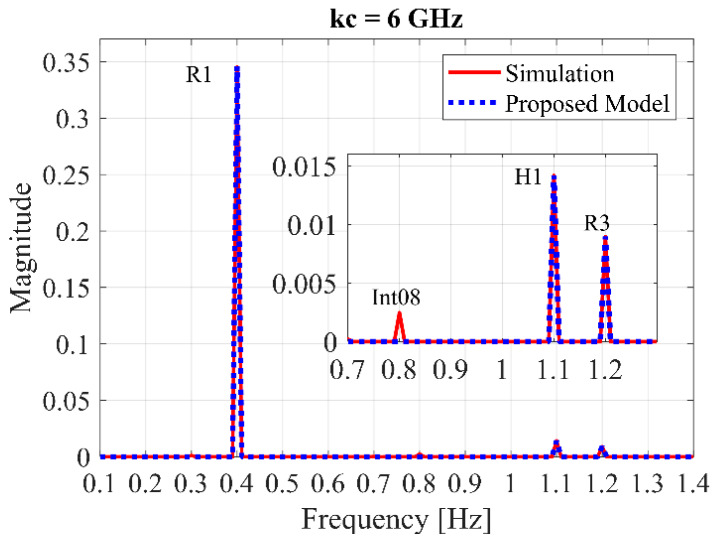
Comparison between the proposed and conventional models. (Insets: zoom in on *H*_1_ and *R*_3_ with scope 0.7–1.3 Hz).

**Figure 5 sensors-20-02916-f005:**
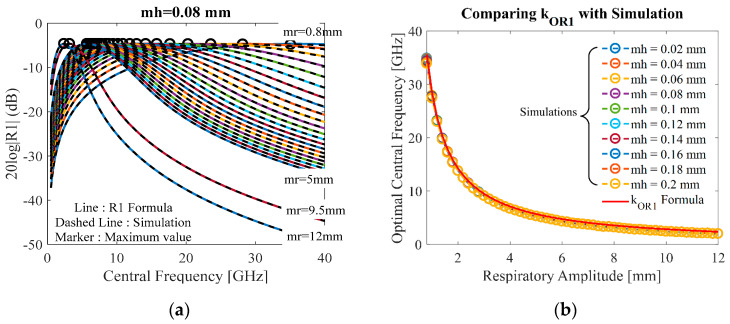
(**a**) Comparison between *R*_1_ model and simulation with small-large respiratory amplitudes m_r_ = 0.8–12 mm at the assumed heartbeat m_h_ = 0.08 mm; (**b**) comparison between *k_OR_*_1_ formula and simulation with varying heartbeat amplitude m_h_ = 0.02–0.2 mm.

**Figure 6 sensors-20-02916-f006:**
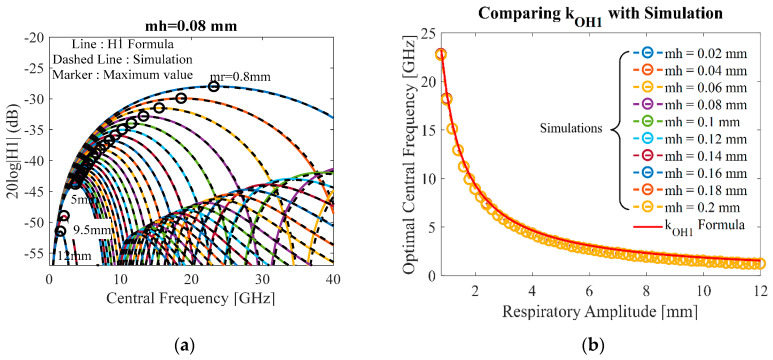
(**a**) Comparison between *H*_1_ formula and simulation with small–large respiratory amplitudes *m_r_* = 0.8–12 mm at the assumed heartbeat *m_h_* = 0.08 mm; (**b**) comparison between *k_OR_*_1_ formula and simulation with varying heartbeat amplitude *m_h_* = 0.02–0.2 mm.

**Figure 7 sensors-20-02916-f007:**
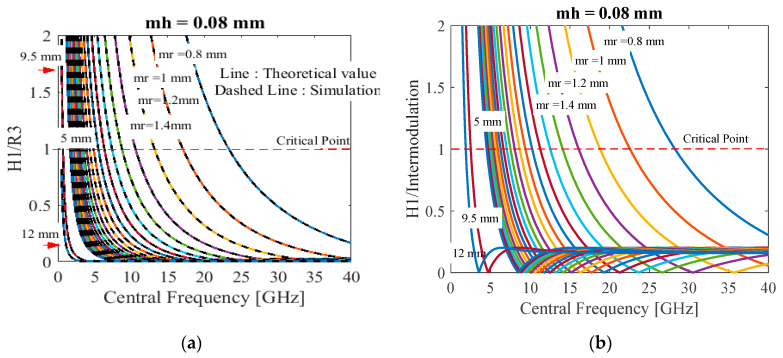
The relative strength versus the central frequency with the small–large respiration amplitudes *m_r_* from 0.8 to 12 mm at the heartbeat *m_h_* of 0.08 mm: (**a**) *H*_1_/*R*_3_; (**b**) *H*_1_/Int08.

**Figure 8 sensors-20-02916-f008:**
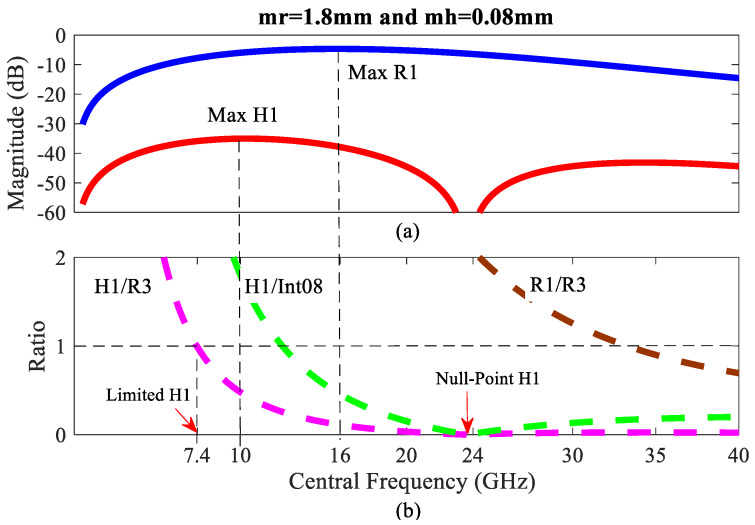
Sequencing the central frequency: (**a**) *R*_1_ and *H*_1_ magnitudes (dB); (**b**) the relative strengths of *H*_1_*/R*_3_, *H*_1_/Int08, and *R*_1_*/R*_3_.

**Figure 9 sensors-20-02916-f009:**
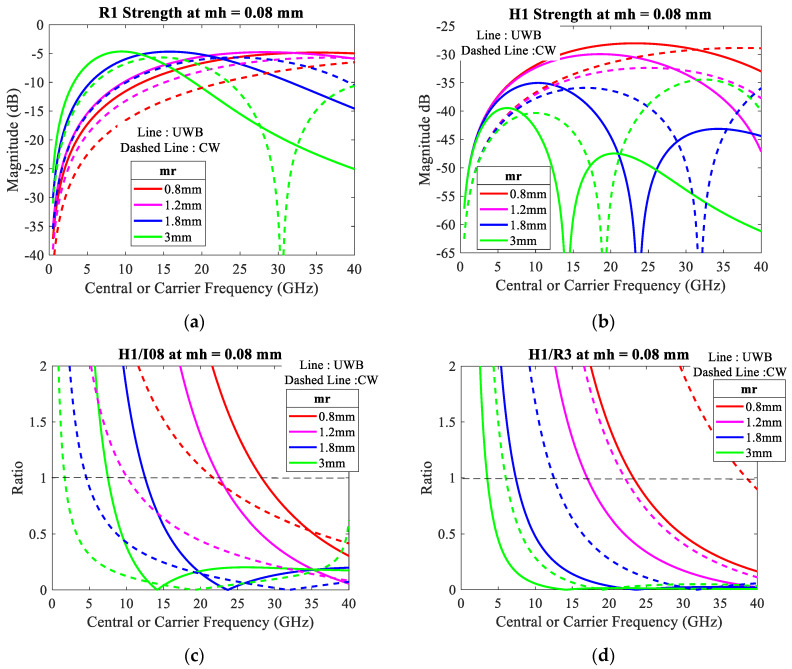
Comparison between ultra-wide band (UWB) central frequency (line) and continuous wave (CW) carrier frequency (dashed line) with respiratory amplitudes mr = 0.8–3 mm at heartbeat amplitude mh = 0.08 mm: (**a**) respiratory strength; (**b**) heartbeat strength; (**c**) the relative strength of heartbeat compared with intermodulation effect of 0.8 Hz; (**d**) the relative strength of heartbeat compared with 3rd respiratory harmonic.

**Figure 10 sensors-20-02916-f010:**
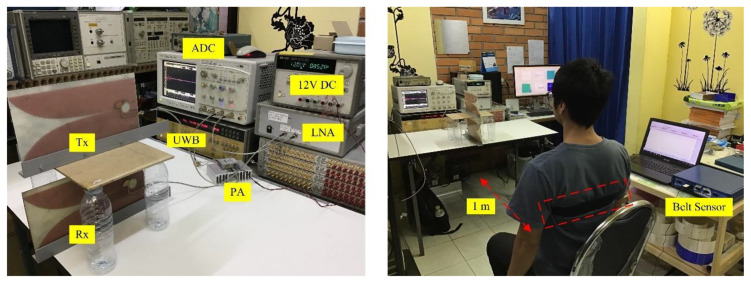
Experimental setup with a human range of 1m according to block diagram in Figure 2a.

**Figure 11 sensors-20-02916-f011:**
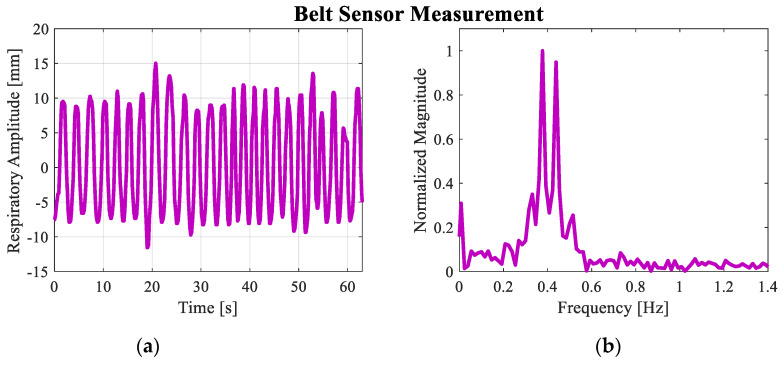
The respiratory movement amplitude measured by a belt sensor; (**a**) time domain; (**b**) frequency domain.

**Figure 12 sensors-20-02916-f012:**
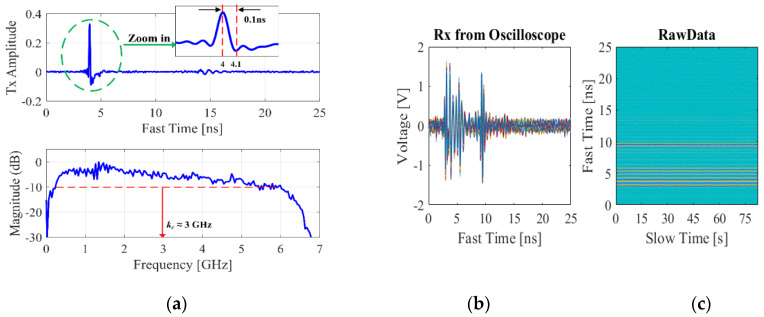
(**a**) The mono-pulse wave form with *t_d_* 0.1ns and 3-GHz central frequency; (**b**) the multiple Rx signals; (**c**) the 2D matrix of the Rx signal data.

**Figure 13 sensors-20-02916-f013:**
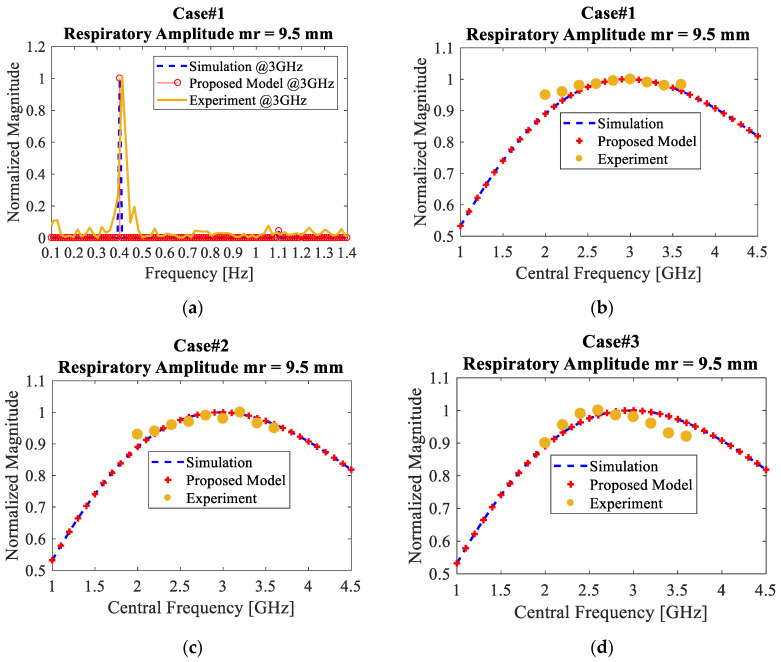
(**a**) The results of the experiment, proposed model, and simulation at 3 GHz central frequency for case#1; the respiration strength versus the central frequency for case #1, case #2 and case #3 shown in (**b**–**d**), respectively.

**Table 1 sensors-20-02916-t001:** Simulation parameters ^1^.

Parameters	Quantity	Value
*d* _0_	nominal distance	1 m
*σ_v_*	reflection amplitude	1
*m_h_*	heartbeat amplitude	0.08 mm
*f_h_*	heartbeat frequency	1.1 Hz (66 beats/min)
*m_r_*	respiratory amplitude	1.8 mm
*f_r_*	respiratory frequency	0.4 Hz (24 beats/min)
*A* _0_	Tx voltage	1 V
*k_c_*	central frequency	6 GHz (follow FCC)
*f_s_fast_*	sampling frequency in fast time	1000 GHz (for comparing to theory)
PRI	pulse repetition interval	25 ns
*τ_max_*	maximum slow time	100 s
*f_s_slow_*	sampling frequency in slow time	200 Hz

^1^ Note all simulations are done with Intel Core i7-3770 CPU of 3.4 GHz, RAM 8 GB, 64-bit, Windows 8.1 Pro, and MATLAB 2018a.

**Table 2 sensors-20-02916-t002:** Equipment List.

Block	Manufacturer	Specifications
UWB source	HP-8133A pulse generator	0.5 V Peak voltage, Central frequency 3 GHz,
Tx and Rx antennas	Vivaldi type (S-band)	2–5 GHz, 10 dBi Angular width (3 dB) ≈ 45°
PA	ZVE-8G + Mini-Circuits	2–8 GHz, 30 dBm
LNA	R&K-AA260-OS	2–5 GHz, 26 dBm
ADC	Agilent Oscilloscope, Infiniium DSO80604B	Max frequency 6 GHz Sampling rate 40 GSa/s
USB port	Agilent GPIB, 82357B	Transfer over 850 KB/sec
Belt sensor	BIOPAC Systems	Records respiratory effort
Transmission Power	-	−5 dBm, bandwidth of 2–5 GHz

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
