# Peer review of "Optimal Central Frequency for Non-Contact Vital Sign Detection Using Monocycle UWB Radar"

_sensors, 2020, doi:10.3390/s20102916_

Round 1
Reviewer 1 Report
My main complaint is about the significance of the presented research. While the theoretical considerations seem sound, the experimental part of the research exposes the impracticality of the proposed model.
In the abstract of the manuscript, the Authors write that “the validity of the formula is verified through simulations and experiments”, but this is not entirely true: the experimental part shows that there is no optimal central frequency because – as shown in figure 13b – the normalized amplitude is almost constant in the investigated range of frequencies. To prove the validity of the model, the range of the central frequencies has to be much wider. Moreover, if the relative strengths of the spectra are considered, there are significant differences between the results of the simulations and the results of the experiments. Most probably, the problem lies – as the Authors noticed – in the hardware imperfections, but then the following question arise: what is the usability of the proposed model? Is the gaussian monopulse a proper model of the transmitted pulse in the real-world applications?
Taking into account the above considerations, I find the conclusions not true and misleading:
- Lines 445–446: “In this section, the optimal central frequency computed by the simple formula (25) can be used to achieve the stronger respiratory strength in non-contact vital sign detection”,
- Lines 455–456: “the measurement results agreed with the formula”.
In my opinion, there are two possible ways to enhance the manuscript before publication:
- The Authors significantly improve the experimental part to actually prove the validity of the proposed model.
- The Authors change the tone of the manuscript, and rephrase the parts implying the practical validity of the model. They have to explicitly say that the model works just in theory, and not in real-world applications. I think the theoretical considerations deserve to be published, but they may be only treated as an illustration that the gaussian monopulse is not a proper model of the pulse in the real-world applications.
In regard to the language, there are some parts of the manuscript that are unintelligible, and should be rephrased:
- Line 36: “UWB pulses are generated a very short time period…”
- Line 45: “… wavelength is sensitive…”
- Line 49: “The conventional vital sign model [22-39] is introduced a new model described…”
- Line 53: “… measurements of experimental…”
- Lines 118–120: “The coefficients Clu not only cause respiration and heartbeat spectrums, but also cause the intermodulation effect between Jl and Ju 120 functions.”
- Line 346: “However, in radar through the wall, the heart rate detection is not important…”
- Lines 377–378: “A computer with GPIB programming controls the transmitter, digitizes the oscilloscope, and computes then displays…”
- Lines 392–393: “Next, the UWB source was set to 3 GHz through the time duration tau=0.1ns of…”
- Lines 409–412: “This section would like to prove the concept of the optimal frequency at using the limited tools, in this experiment, the power on the chest surface is not exceeded 10 W/m2 (permissible exposure limit), the electromagnetic radiation poses no safety threat [45].”
Other comments:
- In Equation (4), the symbols σp, σo and σv are not defined.
- In Equation (8), the symbol Y is not defined.
- In Equation (9), the symbol λ is not defined.
- Line 178: In what sense is the 7th order of power-series “sufficient” to obtain the accurate approximation? Why was the 7th order chosen?
Author Response
Thank you so much for the comments that are helpful in furthering this particular research work, we have attached the response file.

Reviewer 2 Report
1) Authors not clearly defined criterion for optimizing the center frequency
2) There is no analysis of the results obtained, e.g. presented in Figure 5.
3)In my opinion, the article presents a different view on the formula c / 2B (distance resolution). The central frequency is closely related to bandwidth B in monocycle UWB radar.
4) The advantage of this article is to attempt to present mathematical respiratory model and heartbeat model.
5) Table 4 gives the value fs_fast = 1000 GHz.?!
Author Response
Thank you so much for the comments that are helpful in this research work, we have attached the response file.

Reviewer 3 Report
[1] Grammar and writing style need to be improved. Professional editing service is recommended.
[2] Eqn.(1): Suggest to replace “tau” with the Greek “\tau”.
[3] Figure 1: Please redraw the curves and use symbols compatible to those in the text.
[4] Figure 3: Eqn.(10) indicates that the magnitude is proportional to the lth power of frequency, not a linear function of frequency. Please double check.
[5] Line 153: Suggest to change “Clu are proportional to mr, mh and kc” to “Clu increases as mr, mh or kc is increased”
[6] Line 160: Please elaborate why choose l=0.
[7] Line 173: Please elaborate why not choose the third harmonic, the magnitude could be larger than that of the first harmonic.
[8] Line 182: Please specify whether the root, 2πmr/λc = 0.597, is obtained by solving eqn.(24) with numerical root-searching algorithm. If so, why not solve eqn.(23) directly?
[9] Line 191: Please elaborate why choose u=0.
[10] Line 200: Please elaborate why not choose the third harmonic, the magnitude could be larger than that of the first harmonic.
[11] Line 209: Please specify whether the root, 2πmr/λc = 0.3901, is obtained by solving eqn.(24) with numerical root-searching algorithm. If so, why not solve eqn.(31) directly?
[12] Line 237: Please define “null-point”
[13] Line 238: Please elaborate how IQ detection/demodulation is applied in this case.
[14] Figure 4: Please describe the spike labeled “int08” in the inset.
[15] Figure 4: Please elaborate the difference between “simulation” and “proposed model”.
[16] Figure 5(a): Please display onlu five to six curves.
[17] Figure 5(b): Data points are indistinguishable, please use another format to present these results.
[18] Figure 5(c): Suggest to use half number of mh data points.
[19] Figure 6: Please follow the same suggestions on Figure 5.
[20] Figure 7: Please follow the same suggestions on Figure 5.
[21] Figure 8: Please elaborate “I08”.
[22] Figure 9: Suggest to change the horizontal label to “Central or Carrier Frequency (GHz)”
[23] Line 371: The statement “in Figure 9(a), the UWB radar can detect the respiratory strength better than CW at the same frequency” does not apply at higher frequency.
Please rephrase.
[24] Section 5: Suggest to present a few more sets of experiments data.
Author Response
Thank you so much for the comments that are useful in this work, we have attached the response file.

Round 2
Reviewer 1 Report
The Authors have answered all of my comments, and introduced some essential changes to the manuscript; therefore, I recommend accepting the manuscript for publication.
Author Response
Dear Assistant Editor
from now reviewer's comment, the Authors have answered all of my comments, and introduced some essential changes to the manuscript; therefore, I recommend accepting the manuscript for publication.
Reviewer 2 Report
The authors write:
"= we have analyzed them in section 4.1 with references."
The analysis comes to the description of the results of the simulation. In this part, a description of why this is the case, would be recommended. This text would summarize previous analyzes.
Author Response
Dear Professor
Thank you so much for your helpful comments.
Responses to reviewers' comments
1) The analysis comes to the description of the results of the simulation. In this part, a description of why this is the case, would be recommended. This text would summarize previous analyzes.
= We have rewritten that the description in Line 271-287.

Reviewer 3 Report
Previous comments have been well addressed.
Author Response
Dear Assistant Editor
from now reviewer comments, Previous comments have been well addressed.